# Association of Vascular Endothelial Growth Factors (VEGFs) with Recurrent Miscarriage: A Systematic Review of the Literature

**DOI:** 10.3390/ijms24119449

**Published:** 2023-05-29

**Authors:** Nadine Abu-Ghazaleh, Shaun Brennecke, Padma Murthi, Vijaya Karanam

**Affiliations:** 1Department of Maternal-Fetal Medicine, Pregnancy Research Centre, Royal Women’s Hospital, Parkville, VIC 3052, Australia; nadine.abughazaleh@mh.org.au (N.A.-G.);; 2Department of Obstetrics and Gynaecology, University of Melbourne, Parkville, VIC 3052, Australia; 3Department of Pharmacology, Monash Biomedicine Discovery Institute, Monash University, Clayton, VIC 3168, Australia

**Keywords:** recurrent miscarriage, vascular endothelial growth factor, endometrium, placenta, decidua, serum

## Abstract

Recurrent miscarriage (RM) can be defined as two or more consecutive miscarriages before 20 weeks’ gestation. Vascular endothelial growth factors (VEGFs) play an important role in endometrial angiogenesis and decidualization, prerequisites for successful pregnancy outcomes. We conducted a systematic review of the published literature investigating the role of VEGFs in RM. In particular, we explored the methodological inconsistencies between the published reports on this topic. To our knowledge, this is the first systematic literature review to examine the role of VEGFs in RM. Our systematic search followed PRISMA guidelines. Three databases, Medline (Ovid), PubMed, and Embase, were searched. Assessment-bias analyses were conducted using the Joanna Bigger Institute critical appraisal method for case-control studies. Thirteen papers were included in the final analyses. These studies included 677 cases with RM and 724 controls. Endometrial levels of VEGFs were consistently lower in RM cases compared to controls. There were no consistent significant findings with respect to VEGFs levels in decidua, fetoplacental tissues, and serum when RM cases were compared to controls. The interpretation of studies that explored the relationship between VEGFs and RM is hampered by inconsistencies in defining clinical, sampling, and analytical variables. To clarify the association between VEGF and RM in future studies, researchers ideally should use similarly defined clinical groups, similar samples collected in the same way, and laboratory analyses undertaken using the same methods.

## 1. Introduction 

Recurrent miscarriage (RM) is defined as two or more consecutive miscarriages before 20 weeks’ gestation recognized by ultrasound or histopathology [1]. It affects approximately 1–5% of women who conceive. Multiple factors contribute to miscarriages, including fetoplacental chromosomal abnormalities, uterine abnormalities, endocrine disorders, acquired thrombophilia, parental balanced translocations, and immune disorders. However, the underlying causes remain undetermined in 50–70% of patients who experience RM [2].

RM with unknown etiology is defined as unexplained RM or idiopathic RM [1]. Recent advances in cytogenetics and immunogenetics have expanded our knowledge of implantation and maternal–fetal interactions. The placenta at the maternal–fetal interface provides the fetus with the metabolic requirements necessary for development through the exchange of nutrients and wastes [3]. To achieve this, the placenta maintains its own circulation and metabolism via angiogenesis, the formation and remodeling of blood vessels in a vascular network [4]. The three steps of angiogenesis—initiation, proliferation–invasion, and maturation–differentiation—are all critical for normal placental development and successful implantation [5].

There are several well-defined families of angiogenic growth factors that play roles in the development of mature functioning blood vessels. The best characterized is the Vascular Endothelial Growth Factor (VEGF) family (Figure 1). VEGFs are glycoproteins with angiogenic properties [6,7] such as induction of vascular permeability [6,8,9], stimulation of endothelial cell division and migration [9,10] and in vivo angiogenesis [6]. A good endometrial blood supply is usually considered to be a marker for endometrial receptivity [11]. This suggests that placental angiodysplasia and associated vascular endothelial dysfunction may be an important etiology of unexplained RM [12,13].

The direct relationship between VEGFs and RM remains debatable with inconsistent results across the studies. In view of the established crucial role of endometrial factors in facilitating a successful human pregnancy (through angiogenesis and vascular tone reactivity), we conducted a systematic review of the literature to evaluate the role of VEGFs in RM. We also explored the clinical, sampling, and analytical variables in this published literature. To our knowledge, this is the first systematic review of the literature to explore the role of VEGFs in RM.

## 2. Methods and Research Design

### 2.1. Literature Search and Data Extraction 

A systematic literature search following PRIMSA guidelines was devised (Figure 2). Three databases Medline (Ovid), PubMed, and Embase were searched for relevant studies published between the years 2011 and 2022 (Appendix A). Studies were included in the systematic review if they were published in English and complied with the inclusion criteria defined below. The results from the initial search were combined and duplicates were removed using the Endnote X8 and Covidence referencing systems. Citation searching was utilized to augment the initial results. A sample of the search strategy and its result is provided in Appendix A.

Data from included studies were extracted, stored, and analyzed using spreadsheets. Population characteristics and clinicopathological details of the cases and controls were extracted and summarized on excel spreadsheets. Baseline data extracted included country, tissue type utilized, laboratory methodology, statistical analysis, definition and number of cases and controls, and gestational age (Table 1). The aims and results of each study were extracted and are summarized in Table 2. Each study was reviewed by a primary reviewer (NA) and was independently checked for accuracy against the original publication by a second reviewer (VK).

Assessment of bias analyses was conducted using the critical appraisal checklist for studies reporting prevalence data of Joanna Bigger Institute (JBI) (Appendix A)

The Student’s independent samples t-test was conducted using Jamovi Version 2.3.21.0, Sydney, Australia to test the null hypothesis that the serum VEGF level (pg/L) is the same between cases of RM and controls [15].

Serum VEGF data were exported into Review Manager 5.4.1 for quantitative data analysis. The meta-analysis command with random effects to account for heterogeneity was used to estimate the mean difference between serum VEGF in RM cases compared to controls. A quantification of heterogeneity across studies was presented as an I2 score [16].

Almawi et al. was excluded in the meta-analysis and the Student’s independent samples t-test as median values were reported and no raw data were available to calculate the mean [17].

**Table 1 ijms-24-09449-t001:** Patient characteristics and methodologies utilized in the included studies. Information includes the definition of cases and controls, the sample size of cases and controls, patient demographics, laboratory methodologies, and statistical analyses.

Author, Year	VEGF Type/ Tissue Type	Methods	Statistics Analysis	Country/ Ethnicity	Patient Characteristic	Definition of Cases	Case (n)	Definition of Controls	Control (n)	Time of Menstrual Cycle/ Gestational Age and Confirmation Method
Almawi et al., 2013 [17]	Serum VEGF	ELISA	Unpaired student’s t-test for parametric data Pearson x2 or Fisher’s exact test was used to assess intergroup significance Mann–Whitney U-test for VEGF median comparisons	Bahrain/Middle Eastern	Age (cases) mean of 31.6 + 5.4 year Age (control) mean of 31.6 + 4.9 year BMI (cases): 26.3 + 5.4 BMI (control): 25.2 + 4.3	Three or more consecutive spontaneous miscarriages of unknown etiology. Healthy women <40 years old at first. No significant medical history. Women were excluded if screening investigations revealed a possible contributing factor for their pregnancy losses.	296	Multiparous non-pregnant women with at least two live births. No adverse pregnancy outcomes. No personal or family history of RM.	305	First 12 weeks of pregnancy
Amirchaghmaghi et al., 2014 [18]	Serum VEGF	RT-PCR ELISA	T-test or Mann–Whitney	Iran/Middle Eastern	Age (cases) mean of 30.9 ± 1.24 Age (control) mean of 33.8 ± 1.10 BMI (cases): 25.52 ± 0.99 BMI (control): 26.1 ± 2.11	History of recurrent spontaneous miscarriage of unknown etiology. Women were excluded if screening investigations revealed a possible contributing factor for their pregnancy losses.	10	Non-pregnant women with regular menstruation. At least one successful term pregnancy.	6	Weeks’ gestation (cases): Mean of 8.5 ± 1.014
Atalay et al., 2016 [19]	Serum VEGF	ELISA	T-test and non- parametric Mann–Whitney U test were used in analysis between groups	Turkey /Middle Eastern	Age (cases): mean of 28.2 ± 5.2 Age (control) mean of 27.5 ± 4.4 BMI (cases): mean of 26.4 ± 5.1 BMI (control): mean of 25.1 ± 4.3	Three or more consecutive spontaneous miscarriages. Excluded if late 1st trimester or 2nd trimester miscarriages. No significant medical history. Women were excluded if screening investigations revealed a possible contributing factor for their pregnancy losses.	21	Healthy ongoing pregnancies. Conceived spontaneously.	24	Between 5th and 10th week of pregnancy. Transvaginal obstetric ultrasonography for the evaluation of gestational age was conducted at the same day of sample collection.
Bagheri et al., 2013 [20]	Serum VEGF-A and VEGF-C	ELISA	Student’s *t*-test was used for parametric data; Pearson correlation coefficient for the relationship between quantitative variant	India/south Asian of Indian origin.	Age (cases): mean of 30.6667 Age (control/pregnant women): mean of 28. 6429 Age (control/healthy women): mean of 28.9571 BMI (cases): mean of 27.8207 BMI (control/pregnant women): mean of 24.1539 BMI (control/healthy non-pregnant women): mean of 24.9033	Two or more spontaneous miscarriages.	90	Control group 1: Pregnant women, no history of RM, at least one child, women were excluded if screening investigations revealed a possible contributing factor for their pregnancy losses. Control group 2: Healthy non-pregnant women.	Control 1: 70 Control 2: 70	RM: before 20 weeks of pregnancy
Banerjee et al., 2013 [13]	Endometrial biopsy for VEGF	ELISA IHC—Semi-quantitative scoring was performed independently by two observers to assess the staining intensity. A final IHC score was obtained by multiplying intensity score and extent of stained cells.	Unpaired Student’s *t*-test for parametric data	India/south Asian of Indian origin.	Age: cut-off of <35 years BMI: cut-off of ≤28	Three or more consecutive spontaneous miscarriages of unknown etiology. No history of gynaecological disorders. Have not received any medication during the past three months.	66	Parity between 2 and 5 (2.5 ± 0.12). Normal regular menstrual cycles. No history of failed pregnancies or other significant clinical abnormalities.	50	First 12 weeks of pregnancy. Ultrasonography (USG) for serial folliculometry was performed at day 10 onwards in all cases to monitor follicular growth till ovulation occurred. Biopsies performed on day 18–22 of menstrual cycle.
Gupta et al., 2019 [21]	Serum VEGF	ELISA	Not defined	India/south Asian of Indian origin.	Age (cases): 28.9 Age (control): 28.2 BMI (cases): 25.2 BMI (control): 24.0	Three or more consecutive spontaneous miscarriages.	13	At least one successful pregnancy. No personal history of miscarriage.	30	Not defined
He et al., 2016 [22]	Decidua and chorionic villi for VEGF	Western blot qRT-PCR IHC—Semi-quantitative scoring was performed independently by two observers to assess the staining intensity. A final IHC score was obtained by multiplying intensity of staining and percentage of positive staining. The results were generated by averaging the scores of the five views.	Student’s *t*-test for parametric data; Mann–Whitney test was applied for non-parametric data	China/Asian	Age (cases): Mean of 30.3 ± 4.0 Age (control): Mean of 28.7 ± 5.1	Two or more spontaneous miscarriages. Diagnosis of unexplained recurrent miscarriage.	28	Normal early pregnancy with a surgical termination of pregnancy request for unwanted pregnancy. History of prior healthy live births. No abnormal pregnancy history including previous miscarriage, ectopic pregnancy, or still birth.	28	First 12 weeks of pregnancy. Gestational days (cases): Mean of 62.8 ± 8.3. Gestational days (control): Mean of 59.1 ± 7.6. Embryo death confirmed by ultrasound.
Lash et al., 2011 [23]	Endometrial biopsy for VEGF-A, VEGF-C, VEGF-D	qRT-PCR IHC—Semi-quantitative IHC score was obtained by multiplying intensity of staining percentage of cells and staining intensity.	ANOVA post-hoc Fischer unpaired Student’s *t*-test for parametric data	England/Multi-ethnic population	Age (cases): median of 38 (29–44) Age (control): median of 42 (28–51)	Three or more consecutive spontaneous miscarriages. Women were excluded if screening investigations revealed a possible contributing factor for their pregnancy losses.	14	No definition	29	LH + 7 (+2 days)
Pang et al., 2013 [24]	Chorionic villus tissues for VEGF Serum VEGF and VEGF-R1 (Flt)	ELISA IHC—Semi-quantitative IHC score was obtained by determining the percentage of positively stained cells (trophoblastic and interstitial cells) in all fields counted (10 fields for each specimen).		China/Asian	Age (cases): Mean of 33.2 ± 2.9 Age (control): Mean of 32.6 ± 2.1 BMI (cases): median of 20.3 BMI (control): median of 21.2	Two or more spontaneous miscarriages. No history of adverse pregnancy outcomes, including preterm labour, had no current illnesses, did not use regular medication or smoke. Normal chromosome analyses of male and female partners.	32	Normal early pregnancy with a surgical termination of pregnancy request for unwanted pregnancy. No history of spontaneous abortion.	50	Between 6 and 12 weeks of pregnancy. Gestational weeks (cases): 10.9 ± 1.56. Gestational weeks (control): 10.1 ± 1.91. Embryonic death confirmed by ultrasound.
Papamitsou et al., 2021 [25]	Decidua basalis, decidua parietalis and trophoblast for VEGF	IHC—Semi-quantitative IHC score was performed independently by two researchers and obtained by determining the intensity of staining evaluated as negative (–), weak (+), moderate (++), and strong (+++).	Mann–Whitney test.	Greek/European	Age (cases): range of 35–42 Age (control): range of 27–39	Three or more spontaneous miscarriages. Women were excluded if screening investigations revealed a possible contributing factor for their pregnancy losses.	20	Normal early pregnancy with a surgical termination of pregnancy request for unwanted pregnancy.	20	First 12 weeks of pregnancy
Sadekova et al., 2015 [26]	Endometrial VEGF	IHC—Semi-quantitative IHC score was obtained by determining the intensity of staining evaluated on a 0–3 scale.	Mann–Whitney U test, chi square test	Russia/Asia	-	Two or more consecutive spontaneous miscarriages.	24	Fertile women. No history of concomitant gynaecological disease. At least one successful pregnancy.	15	First trimester of pregnancy. Ovulation confirmed by ultrasonography. Endometrial biopsies taken on day 8 after ovulation.
Scarpellini et al., 2019 [27]	Trophoblast Decidua VEGF and VEGFR-1	IHC—Semi-quantitative IHC score was performed independently by two authors and was obtained by multiplying intensity of staining and percentage of cells stained for each intensity.	Mann–Whitney U test	Italy/European	Age: (cases): pregnancy started at 31.6 ± 2.3 Age: (control): pregnancy started at 30.8 ± 2.2 BMI (cases): 27.4 ± 1.9 BMI (control): 27.8 ± 1.8	Women were excluded if screening investigations revealed a possible contributing factor for their pregnancy losses. Pregnancies not obtained by artificial reproductive technology.	15	Normal early pregnancy with a surgical termination of pregnancy request for unwanted pregnancy. No history of previous abortions.	15	Control: 9.4 ± 1.1 (7–10). Cases: 8.1 ± 1.2 (5–9).
Vuorela et al., 2000 [28]	Placental villi, decidua, endometrial glands, invading trophoblast for VEGF	IHC—Semi-quantitative IHC score was performed independently by two observers and was obtained by determining the staining intensity scored from negative (–) to faint (+), medium (++), and strong (+++) staining.	No statistical analysis	Finland/European	Age (cases/ missed abortion): 41.7 ± 1.3 Age (cases/blunted ovum): 29.7 ±1.7 Age (control): 28.6 ± 1.8	Three or more consecutive spontaneous miscarriages. Women were excluded if screening investigations revealed a possible contributing factor for their pregnancy losses.	18	No definition. No miscarriage.	12	Cases/ missed abortion: 9.2 ± 0.3 (7–11) 8. Cases/blunted ovum: 8.3 ± 0.4 (7–10). Control: 9.4 ± 0.4 (7–11)

**Table 2 ijms-24-09449-t002:** Summary of the aims and the statistically significant results of the different included studies.

Author, Year	Aims	Summary of Results
Almawi et al., 2013 [17]	Investigate whether RM is associated with changes in VEGF serum levels. Investigate whether RM is associated with polymorphisms in the VEGFA gene.	The median serum VEGF level was reduced in RM cases compared with control women. Serum VEGF levels correlated with *2460T/C, 398G/A* and *2583T/C* genotypes. Higher minor allele frequency (MAF) and genotype distribution of *2460T/C [corrected P(Pc)1⁄40.003], 398G/A(Pc 1⁄40.016)* and *2583T/C(Pc,0.001)* single nucleotide polymorphisms (SNPs) were seen in RM cases than control women. Increased RM risk was seen with homozygous *2460T/C* and *398G/A* SNPs and with heterozygous *2583T/C*, which had a stronger effect when homozygous.
Atalay et al., 2016 [19]	Investigate whether maternal VEGF levels are associated with RM.	The median serum VEGF level was higher in RM cases compared with control group. Serum VEGF levels did not differ with gestational age within the RM and control groups. A positive correlation was found between VEGF levels and the patients’ age within RM group.
Amirchaghmaghi et al., 2014 [18]	Investigate the serum VEGF concentration in patients with a history of RM compared with normal fertile women.	Serum level of VEGF was higher in RM cases compared with the control group.
Bagheri et al., 2017 [20]	Investigate the relationships between serum level of VEGF-A and VEGF-C with clinical characteristic in women with RM and compare to pregnant and healthy women.	Maternal levels of VEGF-A and VEGF-C were lower in RM cases compared to healthy and pregnant women as control groups. Univariate analysis demonstrated that clinical characteristic factors were associated with concentration of VEGF-A and VEGF-C in cases and controls.
Banerjee et al., 2013 [13]	Identify the significant factor(s) responsible for vascular dysfunction in women with RM during window of implantation.	The angiogenic and vasoactive factors including VEGF, eNOS, NO, and ADM were found to be downregulated and SEBF grossly affected in RM cases endometrium compared to control. Multivariate analysis identified IL-10, followed by VEGF and eNOS as the major factors contributing towards vascular dysfunction in RM women. IL-10, VEGF, and eNOS are strongly correlated with blood flow impairment and endometrial vascular dysfunction.
Gupta et al., 2019 [21]	Identify the level of VEGF in cases of RM and compare it with women with one or more successful term pregnancies. Evaluate the underlying etiology in patients with RM Compare the level of VEGF in women with unexplained RM, and compare it with other underlying causes of RM and to assess if a difference exists between them.	Serum VEGF level was reduced RM cases compared to control. Serum VEGF level was lower in RM cases with underlying aetiology compared to women with unexplained RM.
He et al., 2016 [22]	Investigate whether the expressions of VEGF and Cx43 were altered in the first-trimester tissues (chorionic villi and decidua) collected from women with RM compared to those from healthy early pregnant women.	The immunoreactivity of VEGF in either chorionic villi or decidua was dramatically reduced in RM group compared to the control as revealed by immunostaining, Western blot, and VEGF mRNA.
Lash et al., 2011 [23]	Investigate the temporal and spatial expression of series of angiogenic growth factors (AGFs) and their receptors: vascular endothelial growth factor (VEGF)-A, VEGF-C, VEGF-D, VEGF-R1, VEGF-R2, VEGF-R3, platelet-derived growth factor (PDGF)-BB, PDGF-Ra, PDGF-Rb, transforming growth factor (TGF)-b1, TGF-bRI, TGF-bRII, angiopoietin (Ang)-1, Ang-2 and Tie-2, in the proliferative, early secretory and mid-late secretory phase endometrium from control women as well as in the mid-late secretory phase of women with a history of RM. Four cell types were investigated, namely, glandular epithelium, stromal cells, vascular smooth muscle cells (VSMCs), epithelial cells (ECs).	VEGF-A: moderate VEGF-A immunoreactivity in VSMCs, ECs and glandular epithelial cells across the menstrual cycle that was reduced in RM compared with mid-late secretory controls. VEGF-C: no difference in staining intensity for VEGF-C in RM cases compared to controls in all stages. VEGF-D: no difference in staining intensity for VEGF-D in RM cases compared to controls in all stages. VEGF—R1: immunostaining intensity was increased in all four cell types in women with RM compared with mid-late secretory phase endometrium from controls. VEGF-R2: moderate stromal cell immunoreactivity across the menstrual cycle of control women that was reduced in RM. VEGF-R3: immunoreactivity in glandular epithelial cells was increased in RM compared with mid-late secretory controls.
Pang et al., 2013 [24]	Investigate the levels of sFlt-1 and VEGF in serum and chorionic villus of RM patients compared to control.	The mean seum sFlt-1 level was higher in normal pregnancy group compared to the non-pregnant group. The mean VEGF level was higher in the normal pregnancy compared to the non-pregnant group. The mean sFlt1 concentration was higher in RM cases compared to women with early normal pregnancy. The VEGF concentration was significantly higher in RM cases compared to the normal pregnancy group. The sFlt-1/sFlt-1/VEGF ratio in serum was significantly increased in RM cases compared with normal pregnancy women.
Papamitsou et al., 2021	Investigate the role of VEGF, BCL-2, and BCL-6 in RM.	Increased levels of VEGF, BCL-2, and BCL-6 in RM cases compared to control.
Sadekova et al., 2015 [26]	Investigate the expression of VEGF in the endometrium of women with Luteal phase defect. Investigate VEGF expression in women with RM and fertile women.	VEGF mRNA transcription was decreased in endometrium of women with RM compared to controls. In RM, VEGF mRNA expression in the LPD endometrium was lower than in the endometrium with normal maturity.
Scarpellini et al., 2019 [27]	Investigate the effects of G-CSF treatment on the maternal fetal interface using immunohistochemistry to assess the expression of G-CSF and its receptor, the VEGF and its receptor VEGFR-1, and Foxp3 in the trophoblast and decidua of first trimester miscarriages of RPL women treated with G-CSF that miscarried again despite the treatment, in no treated RPL and in normal first trimester pregnancies.	No differences in the expression of VEGF and VEGFR-1 in the decidua of RM cases and controls. Reduction in the staining for VEGF in the trophoblast, mainly syncytiotrophoblast, of RM cases compared to controls and RM cases treated with G-CSF. Increase in the expression of VEGFR-1 in the trophoblast of RM cases compared to control and RM cases treated with G-CSF.
Vuorela etl al., 2000 [28]	Investigate whether the expression of VEGF, VEGFR-1, -2 or -3 or the Tie-1 or Tie- 2 receptors in the placenta or decidua are altered in RM compared to control.	Compared with controls, the MA and BO groups showed: ○ Diminished placental trophoblastic VEGF immunoreactivity; ○ Weaker VEGFR-1 and -2 immunoreactivity in decidual vascular endothelium; ○ Reduced placental trophoblastic Tie-1 receptor immunoreactivity; ○ Reduced decidual vascular endothelial Tie-1 and -2 receptor immunoreactivity. The absence of VEGFR-3 immunoreactivity in decidual vascular endothelium was noted in all study groups. Placental villi from the BO group presented blood vessel-like structures positive for VEGF, VEGFR-1, -2, -3, Tie-1 and Tie-2 receptor.

### 2.2. Inclusion Criteria

Population

All case-control studies (regardless of sample size) that investigated the association of RM and VEGFs in the endometrium, decidua, fetoplacental tissue (including chorion and trophoblasts), and/or maternal serum levels were included. Cases were excluded if a history of recurrent idiopathic miscarriage was unclear. No limitations were placed on age, ethnicity, country of origin, family history of recurrent miscarriage, or any other patient demographics. 

Laboratory Methods

All studies that used laboratory methods for identifying VEGFs’ distribution, localization, concentration, or level in the endometrium, decidua, fetoplacental tissue, or maternal serum were included.

Laboratory methods across studies included the following:(1)Enzyme-Linked Immunosorbent Assay (ELISA) to measure serum levels of the VEGFs.(2)Real-Time Quantitative Reverse Transcription PCR (qRT-PCR) to detect VEGF-gene expression.(3)Immunohistochemistry (IHC) of endometrial, decidua, and/or placental tissue to determine localization and distribution of VEGFs.(4)Western blot in conjunction with IHC and/or qRT-PCR, of endometrial, decidua, or fetoplacental tissue to determine localization, distribution, and levels of VEGFs.

The different laboratory methodologies and tissue types utilized across the studies are summarized in Figure 3.

**Figure 3 ijms-24-09449-f003:**
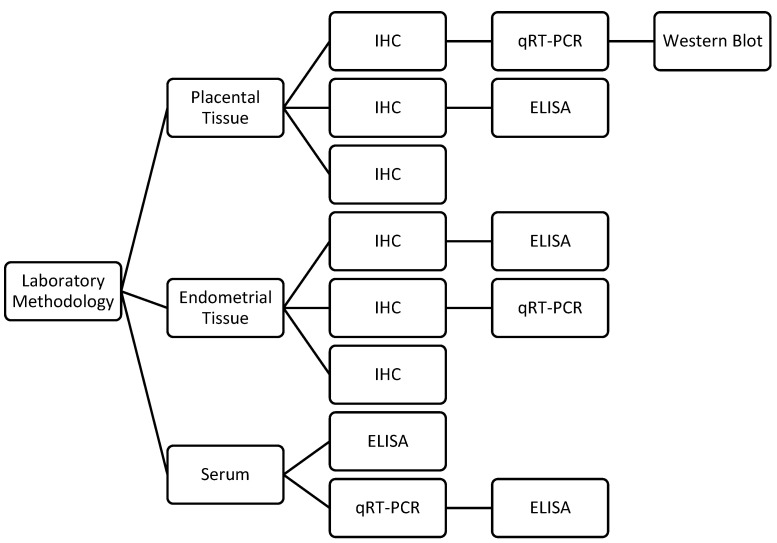
The different laboratory methodologies and tissue types utilized across the studies.

Outcome

All studies with a primary outcome measure of identifying VEGFs in RM were included, provided the definition of RM was two or more miscarriages in the first 20 weeks’ gestation.

### 2.3. Exclusion Criteria

If the definition of RM was not two or more miscarriages in the first 20 weeks’ gestation, the study was excluded.

Studies that recruited cases with other types of miscarriages such as threatened, missed, incomplete, or complete miscarriage were excluded if the cases were not diagnosed with RM.

Studies that used only Western blot investigation of endometrial, decidua and/or fetoplacental tissue were excluded.

## 3. Results and Discussion

### 3.1. Search Results

A total of 480 titles and abstracts were retrieved from the literature searches and an additional 3 from searching reference lists and systematic reviews. Eighteen citations were fully assessed (*n* =18). From these, seven were excluded (*n* = 7) (Figure 2). Appendix A provides detailed information on why each study was excluded.

Thirteen papers were included for the final analyses (*n* = 13). These studies included 677 cases with RM. The studies were across 10 countries. India (*n* = 3) followed by China (*n* = 2) had the largest number of eligible studies. There were many studies published in the European continent (*n* = 3) and Middle Eastern countries (*n* = 3).

The aim and results of each study are summarized in Table 2.

### 3.2. Terms and Definitions

After a full-text review of the included studies, the following presents the definitions of recurrent idiopathic miscarriage, healthy controls, and VEGFs as defined by their authors.

Recurrent Idiopathic Miscarriage

Recurrent miscarriage was defined as two or more miscarriages in the first 20 weeks’ gestation [1]. Unexplained or idiopathic miscarriages were defined as pregnancy loss with ‘known’ causes being excluded. These causes included anatomical abnormalities of the genital tract, chromosome abnormalities of partners, and hematological, endocrinological or immunological risk factors for RM.

Recurrent idiopathic miscarriages also included keywords: pregnancy loss, spontaneous abortion, and vaginal expulsion of fetus.

The different definitions for RM cases are summarized in Table 1.

Healthy Controls

Depending on the study design, definitions of healthy controls included:(1)Pregnant women with no current and/or previous pregnancy complications request a surgical termination of pregnancy for unwanted pregnancy.(2)Pregnant women with no personal and/or family history of miscarriage requesting a surgical termination of pregnancy for unwanted pregnancy.(3)Non-pregnant healthy multiparous women for endometrial sampling.

The different definitions of healthy controls are summarized in Table 1.

Vascular Endothelial Growth Factors

VEGFs comprise VEGF-A, placental growth factor (PlGF), VEGF-B, VEGF-C and VEGF-D, their receptors VEGFR-1 (also called FLT-1), VEGFR-2 and VEGFR-3 (FLT-4), and their co-receptors, neuropilin-1 (NRP-1) and NRP-2 [29].

### 3.3. VEGFs in Endometrial Tissue

Three studies investigated the expression of VEGF in endometrial tissue in cases of RM compared to controls. Banerjee et al. (2013) [13] found that VEGF is significantly downregulated in the endometrial of women with RM. The study also reported that VEGF is significantly associated with vascular dysfunction and blood flow impairment. Lash et al. (2011) found that only VEGF-A expression was reduced in women with RM compared to mid-late secretory controls, whereas there was no difference in staining intensity for VEGF-C and VEGF-D in RM cases compared to controls in all stages of the menstrual cycle. VEGF-R1 and VEGF-R3 were increased in women with RM compared with mid-late secretory phase endometrium from controls. However, stromal cell immunoreactivity for VEGF-R2 across the menstrual cycle was reduced in women with RM compared to control women. Sadekova et al. (2015) [26] reported that VEGF mRNA transcription was decreased in the endometrium of women with RM compared to control.

### 3.4. VEGFs in the Maternal Decidua and Placental Tissues

Five studies investigated the expression of VEGF in the placental tissues in RM compared with controls. He et al. [22] reported that VEGF expression in either chorionic villi or decidua was significantly reduced in the RM group compared with control as revealed by immunostaining, Western blot, and VEGF mRNA [22]. However, Pang et al. reported that RM patients had a higher expression of VEGF in chorionic villi compared with normal pregnancy controls [24]. Papamitsou et al. [25] reported a significantly increased level of VEGF in the decidua and in the chorionic villi in the RM group compared with control. Scarpellini et al. demonstrated no differences in the expression of VEGF and VEGFR-1 in the decidua, a significantly reduced staining for VEGF in the trophoblasts, and a significantly increased expression of VEGFR-1 in the trophoblasts of RM compared with control [27].

### 3.5. VEGFs in Serum

Six studies investigated serum levels of VEGF in RM cases compared with controls. Three studies reported that serum VEGF levels were significantly reduced in RM cases compared with controls and three studies reported that serum VEGF levels were significantly higher in RM cases compared with controls. Bagheri et al. (2017) demonstrated that the concentrations of VEGF-A and VEGF-C were correlated with the clinical characteristics of cases and controls [20]. Gupta et al. reported that serum VEGF level was lower in women with underlying etiology compared to women with unexplained RPL [21]. Atalay et al. reported that serum VEGF levels did not differ with gestational age, both in RM and control groups [19].

An independent *t*-test showed no evidence that the mean serum VEGF level (pg/mL) in cases with recurrent miscarriage is different from the control. The mean serum VEGF levels (pg/mL) in control was 160.94 pg/mL compared to the mean serum VEGF levels in cases of 148.66 (t(8) = 0.21, *p* = 0.84). (Figure 4).

**Figure 4 ijms-24-09449-f004:**
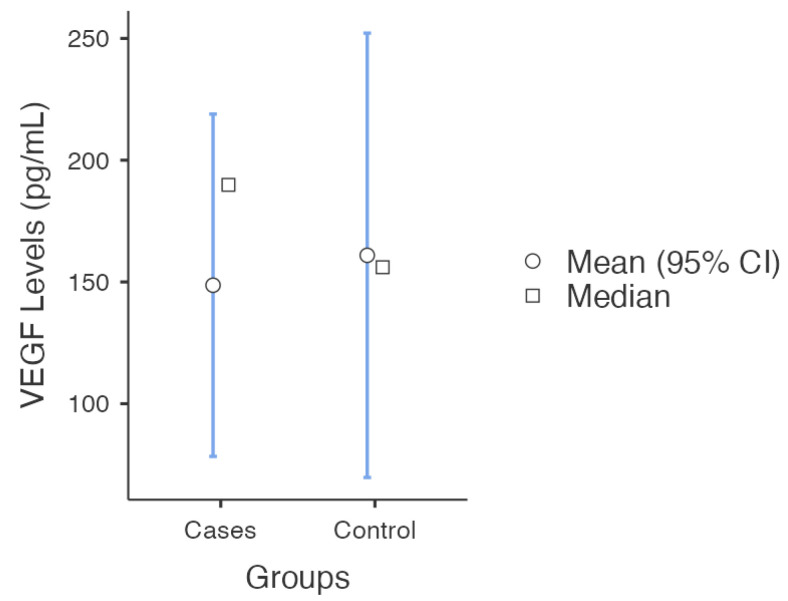
This figure demonstrates the mean serum level of VEGF (pg/mL) in cases compared to control across the different studies.

Upon meta-analysis, the overall mean difference between serum VEGF was 12.71 pg/mL. Levels were higher in controls compared to cases of RM (confidence interval (CI) 95% − 58.7–33.27%, I2:93%). This data was not statistically significant and had a high level of heterogeneity (Figure 5).

**Figure 5 ijms-24-09449-f005:**
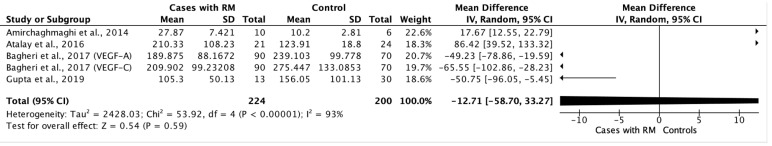
Forest plot of comparison of VEGF levels (pg/mL) in cases of RM with control using a random effect model. (Refs. [18,19,20,21]).

### 3.6. Quality Assessment

Upon quality assessment, a few sources of publication bias were identified based on the JBI criteria. Most studies were of high quality and estimated to have a low risk of bias. Hence, a decision was made not to exclude any studies following the quality assessment. Specifically, 38.46% of the studies (*n* = 5) were assessed as having a low risk of bias on all 10 criteria according to the JBI method, and 23.8% of the studies (*n* = 3) did not address 1 criterion. Areas of concern across the studies that may affect the cumulative evidence included not ensuring that the RM cases and control were comparable in patients’ characteristics and matched appropriately, not providing clear definitions for RM cases and controls, and not identifying confounding factors and strategies used to deal with them. Other areas included not utilizing IHC appropriately, i.e., not using more than one researcher to interpret staining intensity results and the lack of interpretation of a sufficient number of fields. Since qualitative assessment for selection bias was conducted after the papers met the inclusion criteria, no further analysis of bias was warranted (Appendix A).

### 3.7. Discussion

The development of an embryo needs extensive and systematic blood vessels to support implantation and placentation. It has been hypothesized that this is impaired in RM patients. VEGF plays a significant role during implantation by stimulating endothelial cell proliferation and increasing vascular permeability [30,31,32,33]. Although evidence supports an association between VEGF and angiogenesis, the underlying role of VEGF in RM is unclear. Significantly lower vascular, stromal, and glandular expression patterns of endometrial VEGF were observed in RM compared to controls [13,23,26]. However, there were VEGF result inconsistencies when investigating the relationship using decidua, fetoplacental tissues, and serum.

The inconsistencies in the results are potentially attributed to dissimilar methodologies and study designs. To explore the relationship between VEGF and RM, studies should have a consistent design, similar clinical definitions and study populations, and comparable methodologies, together with sufficiently large sample sizes for statistically valid conclusions to be drawn. Table 3, Table 4 and Table 5 identify the clinical, sampling, and analytical variables that must be clearly accounted for when designing a case-control study investigating the role of placental, decidua, endometrial, and serum VEGF and RM.

#### 3.7.1. Clinical Variables

When exploring the relationship between VEGF in RM cases compared to controls, it is essential that both groups are similar, barring the presence of RM in cases and its absence in controls. Despite some studies matching cases to controls based on multiple parameters, the majority neglected essential parameters including BMI, ethnicity, gestation age, menstrual phase, and parity and gravity. Multiple confounding factors including sperm quality, concomitant gynecological diseases, and medical predisposition for RM were not thoroughly explored in the included papers.

Serum, endometrial, placental, and decidual VEGF levels are dependent on multiple factors that might have confounded the results and led to different findings in RM compared to controls. For example, VEGF-gene polymorphisms, which are influenced by ethnicity, might be a factor contributing to the discrepancy in the results [17,34]. Almawi et al. reported a differential association of VEGF variants with RM based on the VEGF genotype, highlighting the contribution of ethnicity and the need for genetic association studies [17]. Bagheri et al.’s univariate analysis also demonstrated that clinical characteristics of both cases and controls were significantly associated with the concentrations of VEGF-A and VEGF-C, which are known to play a relevant role in angiogenesis regulation [20]. Moreover, supporting the influence of patient medical comorbidities on VEGF serum levels, Gupta et al. reported that VEGF levels were lowest in patients with an underlying etiology of miscarriage including hypothyroidism, uterine anatomical abnormalities, and antiphospholipid syndrome compared to patients with unexplained RM [21]. Hence, large sample-size studies, with well-documented clinical variables of cases and controls, are needed to explore the differential expression of VEGF and its relationship with RM (Table 3).

#### 3.7.2. Sampling Variables

When exploring sampling variables, multiple factors could have impacted the perceived association between VEGF and RM (Table 4). A minority of the studies specified the histological layer of uterine, placental, and decidua tissue and the region of sampling. The location of samples within the placental tissue should also be clearly stated as gene expression may vary in central and peripheral placental tissue. The sample timing following the miscarriage/termination of pregnancy should also be well documented. No effect has been reported, in the literature, of this delay in tissue evacuation or “retention time” and the vascular parameters; however, it might influence interpretations of the histopathological observations.

#### 3.7.3. Analytical Variables

Analyses of the serum, endometrial, placental, and decidual tissue should be similar between cases and controls (Table 5). Tissue samples should be correlated with hysteroscopic and/or ultrasonic findings, and samples should be taken from well-defined menstrual cycle phases or gestation ages. Isoforms of VEGFs sequenced, and antibody types used in ELISAs, Western blot, and/or IHC should also be well described in the study reports. The analysis of immunostaining should control for any systemic errors by the utilization of blinding, analysis of the intensity of staining, and analysis of the percentage of cells for each staining intensity.

To explore the association between VEGFs and RM, it is worth noting that multiple angiogenic and immunogenic factors influence endometrial and placental tissue. There are key angiogenesis pathways that could be contributing to this process and overall angiogenic balance of the placenta including TGF-beta (Tumor Growth Factor Beta), FKBPL (FK506-binding protein like), CD44, HSP90 (Heat shock protein 90), and Notch. In the context of placenta development, CD44 and VEGF may be involved in a complex signaling network that regulates placental development. Specifically, CD44 has been shown to modulate the expression and activity of VEGFR1 and its ligands, such as VEGF, in trophoblasts. CD44 has also been shown to regulate trophoblast migration and invasion by modulating VEGF signaling. Moreover, CD44 has been suggested to function as a co-receptor for VEGFR2, another VEGF receptor, and to enhance VEGF-mediated signaling in endothelial cells. This suggests that CD44 may play a role in the formation and maintenance of the placental vasculature [35]. FK506 inhibits migration of human microvascular endothelial cells through binding to the CD44 receptor, leading to downstream effects on the actin cytoskeleton and alterations in expression of CD44 and related proteins, suggesting a co-regulatory pathway between FKBPL and CD44 [14,36]. In placenta development, HSP90 has been shown to interact with and stabilize VEGF, allowing for proper VEGF signaling and angiogenesis. HSP90 also plays a role in regulating the expression of VEGF receptors, such as VEGFR2, on endothelial cells. This interaction between HSP90 and VEGF is important for the development and maintenance of the fetal vasculature and ensuring adequate blood flow [37]. Studies have shown that VEGF induces the expression of Notch signaling components in endothelial cells, which, in turn, promotes the formation of new blood vessels in the placenta. In addition, Notch signaling has been shown to regulate the expression of VEGF receptors in endothelial cells, which is required for the proper response of endothelial cells to VEGF stimulation [38,39]. During placenta development, TGF-beta signaling regulates the differentiation and migration of trophoblast cells, which are specialized cells that form the placenta. It also regulates the remodeling of maternal blood vessels that supply nutrients and oxygen to the developing fetus [38,39]. TGF-beta works in conjunction with VEGF to regulate angiogenesis and blood vessel formation in the placenta [38,39]. The exact interactions of these cytokines, immune cells, and angiogenic factors are not well-defined in the literature. Hence, it is simplistic to attribute the cause of idiopathic RM to one angiogenic factor family, VEGFs. Future studies could attempt to explore the role of VEGF amongst other angiogenic and immunogenic factors that have established or proposed associations with RM.

Finally, this systematic review had limitations in conducting a meta-analysis. One study calculated the median serum VEGF levels as opposed to the mean and hence was excluded from the quantitative analysis [17]. Some studies investigated VEGF and others specifically investigated VEGF-A or VEGF-C and hence a quantitative comparison is not completely accurate. In terms of measuring levels of VEGF using IHC, most studies did not include raw numbers of cases and controls that fell into each category of strength and intensity of staining, which made it difficult to quantitively analyze the data. The heterogeneity was high; therefore, some other confounders such as population characteristics within each study need to be assessed to potentially explain the variability found in this meta-analysis. Ideally, individual participant data (IPD) meta-analysis is needed to obtain more accurate estimates. Despite these limitations, our results yield important conclusions and approaches to assess VEGF levels in future studies.

## 4. Conclusions

Although there is evidence supporting the role of VEGFs in implantation and placentation, the underlying VEGF role in RM is unclear. The overall interpretation of studies that explored the relationship between VEGFs and RM is hampered by inconsistencies in defining clinical characteristics of cases and controls, tissue sampling methods, and technical variations in VEGF identification and measurement. Clinical, sampling, and analytical variables must be well-defined and further multi-ethnic population-based studies of the association between VEGFs and RM are warranted. In this paper, we identified potential study variables that would impact the observed relationship between VEGF and RM, and accordingly, provide recommendations for case-control studies attempting to explore this association.

## Figures and Tables

**Figure 1 ijms-24-09449-f001:**
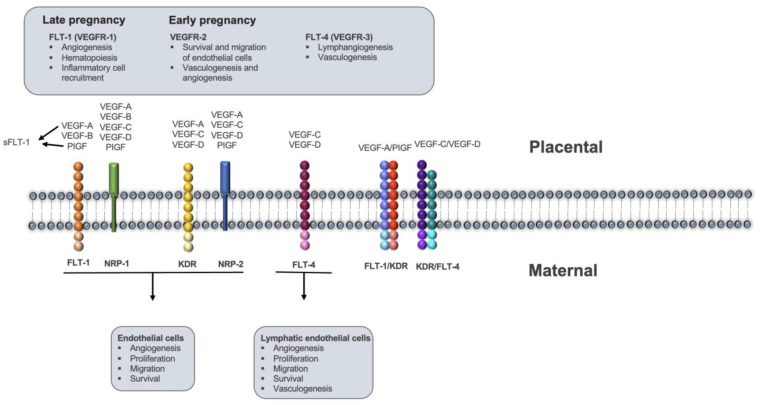
The VEGF family of angiogenic growth factors and receptors: interactions and functions. VEGF-A is a gene found on chromosome 6p12-p21.1 that codes for six different isoforms of VEGF-A, including the predominant isoform VEGF-A165, which binds to the receptors FLT-1, KDR, NRP-1, and NRP-2 and promotes angiogenesis, endothelial cell growth, and vascular permeability. PlGF, which has 42% amino acid sequence identity with VEGF-A, is predominantly expressed in the placenta, heart, and lungs, and plays a role in regulating VEGF-dependent angiogenesis under pathological conditions. VEGF-B, which forms stable heterodimers with VEGF-A, binds to the receptors FLT-1 and NRP-1 and behaves as an endothelial cell mitogen. VEGF-C regulates the lymphatic system during embryogenesis and adult life, while VEGF-D plays a role in lymphangiogenesis and can also activate the receptors FLT-4 and KDR. FLT-1 and KDR are receptor tyrosine kinases expressed on vascular endothelial cells that bind to VEGF-A, VEGF-B, and PlGF, while NRP-1 and NRP-2 are co-receptors that enhance the binding of VEGF-A and PlGF to their respective receptors.

**Figure 2 ijms-24-09449-f002:**
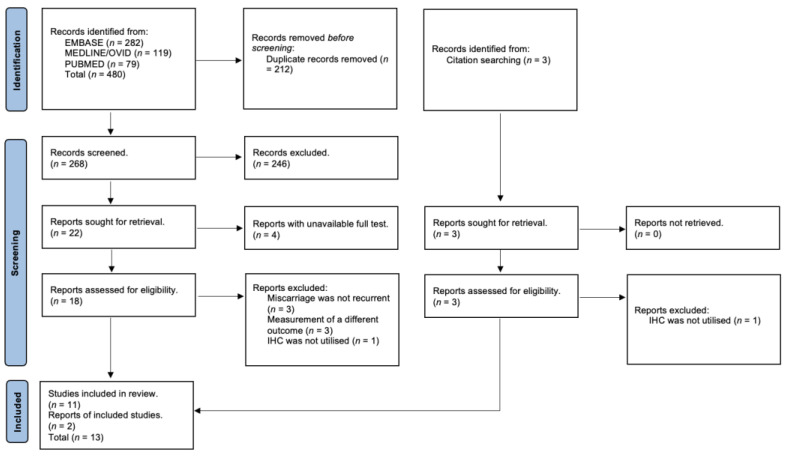
PRISMA chart [14].

**Table 3 ijms-24-09449-t003:** The clinical, sampling, and analytical variables to identify when designing a case-control study investigating the role of endometrial VEGF and RM.

Endometrial Vascular Endothelial Growth Factor		
Clinical Variables for Cases and Controls	Sampling Variables	Analysis Variables
Age	Days following the last pregnancy	Correlation of tissue sample with ultrasonic or hysteroscopic features
BMI	Endometrial sampling techniques	Method of confirming menstrual phase
Ethnicity	Endometrial thickness	Criteria for endometrial histological dating
Parity and Gravity	Anatomical uterine location	Methodology of tissue fixation
Stage, day, and regularity of the menstrual cycle: menstrual phase, secretory phase, proliferative phase.	Histological uterine layer: uterine stratum compactum, stratum spongiosum and/or stratum basalis.	Methodology of endometrial analysis
Assessment of sperm DNA, meiotic alternations, and parameters	Size of tissue fragments	Isoform of VEGF sequenced
History of RM	Number of endometrial glands and presence of stroma	Analysis of immunostaining: utilisation of blinding, analysis of intensity of staining, analysis of the percentage of cells for each staining intensity
History of concomitant gynaecological diseases	Viability of the endometrial sampling	
Medical predispositions for RM *		

* including antiphospholipid syndrome [lupus anticoagulant tested for using the dilute Russell viper venom time and immunoglobulin G (IgG) and IgM anticardiolipin antibodies], thrombophilia (activated protein C resistance, Leiden factor V mutation, prothrombin gene mutation, protein C and S deficiency and antithrombin III deficiency), uterine anomaly (transvaginal ultrasonography), polycystic ovarian syndrome (transvaginal ultrasonography), diabetes (fasting blood glucose), abnormal thyroid function tests or parental balanced translocations (leucocyte culture), TORCH (Toxoplasmosis, Rubella, Cytomegalovirus and Herpes) infections, abnormal paternal and maternal chromosomal analysis.

**Table 4 ijms-24-09449-t004:** The clinical, sampling, and analytical variables to identify when designing a case-control study investigating the role of placental and decidua VEGF and RM.

Placental and Decidua Vascular Endothelial Growth Factor		
Clinical Variables for Cases and Controls	Sampling Variables	Analysis Variables
Age	Type of tissue Placental villi Cytotrophoblasts Syncytiotrophoblast Stromal cells Vascular endothelium Decidua Stromal cells Vascular endothelium	Correlation of gestation age with ultrasonic or hysteroscopic features
BMI	Sampling techniques	Method of confirming termination of pregnancy
Ethnicity	Time of sampling following miscarriage/termination of pregnancy	Criteria of confirming gestation age
Parity and Gravity	Region of the sample analysed: central vs. peripheral	Methodology of tissue fixation
Gestation Age	Viability of the tissue sampled	Methodology of placental and decidua analysis
Assessment of sperm DNA, meiotic alternations, and parameters		Isoform of VEGF sequenced
History of RM		Analysis of immunostaining: utilisation of blinding, analysis of intensity of staining, analysis of the percentage of cells for each staining intensity
History of concomitant gynaecological diseases		
Medical predispositions for RM *		

* including antiphospholipid syndrome [lupus anticoagulant tested for using the dilute Russell viper venom time and immunoglobulin G (IgG) and IgM anticardiolipin antibodies], thrombophilia (activated protein C resistance, Leiden factor V mutation, prothrombin gene mutation, protein C and S deficiency and antithrombin III deficiency), uterine anomaly (transvaginal ultrasonography), polycystic ovarian syndrome (transvaginal ultrasonography), diabetes (fasting blood glucose), abnormal thyroid function tests or parental balanced translocations (leucocyte culture), TORCH (Toxoplasmosis, Rubella, Cytomegalovirus and Herpes) infections, abnormal paternal and maternal chromosomal analysis.

**Table 5 ijms-24-09449-t005:** The clinical, sampling, and analytical variables to identify when designing a case-control study investigating the role of serum VEGFs and RM.

Serum Vascular Endothelial Growth Factor		
Clinical Variables for Cases and Controls	Sampling Variables	Analytical Variables
Age	Type of sample: serum vs. plasma	Correlation of gestation age with ultrasonic or hysteroscopic features
BMI	Time of sampling following miscarriage/termination of pregnancy	Method of confirming termination of pregnancy
Ethnicity	Viability of the tissue sampled	Criteria of confirming gestation age
Parity and Gravity		Methodology of analysis
Gestation Age		VEGF isoforms
Assessment of sperm DNA, meiotic alternations, and parameters		
History of RM		
History of concomitant gynaecological diseases		
Medical predispositions for RM *		

* including antiphospholipid syndrome [lupus anticoagulant tested for using the dilute Russell viper venom time and immunoglobulin G (IgG) and IgM anticardiolipin antibodies], thrombophilia (activated protein C resistance, Leiden factor V mutation, prothrombin gene mutation, protein C and S deficiency and antithrombin III deficiency), uterine anomaly (transvaginal ultrasonography), polycystic ovarian syndrome (transvaginal ultrasonography), diabetes (fasting blood glucose), abnormal thyroid function tests or parental balanced translocations (leucocyte culture), TORCH (Toxoplasmosis, Rubella, Cytomegalovirus and Herpes) infections, abnormal paternal and maternal chromosomal analysis.

## Data Availability

Not applicable.

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
