# Peer review of "Association of Vascular Endothelial Growth Factors (VEGFs) with Recurrent Miscarriage: A Systematic Review of the Literature"

_ijms, 2023, doi:10.3390/ijms24119449_

Round 1

Reviewer 1 Report

This interesting systematic review describes the role of VEGFs in recurrent miscarriage, which is an area of conflicting and inconsistent results.

Generally, a very well written manuscript with informative tables.

My main question is: Why did the authors not perform meta-analyses? I understand that there is variation in many aspects of the studies however this will be taken into the account when meta-analysis is performed. This way rather than the studies being critically appraised through narrative, appropriate data and statistical analysis will be carried out for systemic reviews. This could be done by the use of diagnostic evaluation of VEGFs in RM (true positive/negative and false positive/negative numbers; diagnostic odds ratios) or even through a simple comparison of the means? This can be done in R or RevMan. The authors certainly have enough studies to pool data together for a particular type of VEGF (e.g. for serum VEGF).

Another minor suggestion I have is to discuss other key angiogenesis pathways that could be contributing to this process and overall angiogenic balance of the placenta including TGF-beta, FKBPL, CD44, HSP90, Notch etc.

The authors should acknowledge the fact that other factors contribute to angiogenesis except for the angiogenic growth factors such as metabolites/amino acids, trophoblast function (in terms of remodelling) and stromal cells. 

Author Response

Response to Reviewer 1 Comments

Point 1: My main question is: Why did the authors not perform meta-analyses? I understand that there is variation in many aspects of the studies however this will be taken into account when meta-analysis is performed. This way rather than the studies being critically appraised through narrative, appropriate data and statistical analysis will be carried out for systemic reviews. This could be done by the use of diagnostic evaluation of VEGFs in RM (true positive/negative and false positive/negative numbers; diagnostic odds ratios) or even through a simple comparison of the means? This can be done in R or RevMan. The authors certainly have enough studies to pool data together for a particular type of VEGF (e.g. for serum VEGF).

Thank you for your feedback. We agree that a meta-analysis would have ideal. However, these are the challenges we faced, as discussed with a statistician, when we tried to conduct a meta-analysis.

Serum VEGF

Some studies calculated the median serum VEGF levels and others calculated the mean levels

No individual levels of serum VEGF levels for each patient were given

Some studies investigated VEGF and others specifically investigated VEGF-A or VEGF-C and hence a quantitative comparison is not applicable/not accurate.

Levels of VEGF on IHC

Most studies did not include raw numbers of cases and controls that fell into each category of strength of staining. Initially, we were planning on classifying the staining levels into negative, weak, moderate, and strong and utilising simple statistical analysis to determine whether there a significant difference between cases and controls. We have attempted to contact all authors and ask for raw numbers of how many cases (RM) and controls fell in each category staining. We also wanted the number of cases and controls classified into intensity VEGF expression level as “low” vs “high”. However, none of the corresponding authors got back to us.

Point 2: Another minor suggestion I have is to discuss other key angiogenesis pathways that could be contributing to this process and overall angiogenic balance of the placenta including TGF-beta, FKBPL, CD44, HSP90, Notch etc.

The authors should acknowledge the fact that other factors contribute to angiogenesis except for the angiogenic growth factors such as metabolites/amino acids, trophoblast function (in terms of remodelling) and stromal cells. 

Thank you for your feedback. We have adjusted the manuscript accordingly and included this paragraph on lines (287 – 311)

“To explore the association between VEGFs and RM, it is worth noting that multiple angiogenic and immunogenic factors influence endometrial and placental tissue. There are key angiogenesis pathways that could be contributing to this process and overall angiogenic balance of the placenta including TGF-beta (Tumour Growth Factor Beta), FKBPL (FK506-binding protein like), CD44, HSP90 (Heat shock protein 90), and Notch. In the context of placenta development, FKBPL has been shown to inhibit VEGF signalling by binding VEGFR2 and preventing its activation as well as downregulating the expression of PIGF. This leads to a reduction in the production of new blood vessels and the growth of existing blood vessels in the placenta (1). On the other hand, studies have shown that CD44 and VEGF may be involved in a complex signalling network that regulates placental development. Specifically, CD44 has been shown to modulate the expression and activity of VEGFR1 and its ligands, such as VEGF, in trophoblasts. CD44 has also been shown to regulate trophoblast migration and invasion by modulating VEGF signalling. Moreover, CD44 has been suggested to function as a co-receptor for VEGFR2, another VEGF receptor, and to enhance VEGF-mediated signalling in endothelial cells. This suggests that CD44 may play a role in the formation and maintenance of the placental vasculature (2). In placenta development, HSP90 has been shown to interact with and stabilize VEGF, allowing for proper VEGF signalling and angiogenesis. HSP90 also plays a role in regulating the expression of VEGF receptors, such as VEGFR2, on endothelial cells. This interaction between HSP90 and VEGF is important for the development and maintenance of the fetal vasculature and ensuring adequate blood flow (3). Studies have shown that VEGF induces the expression of Notch signalling components in endothelial cells, which, in turn, promotes the formation of new blood vessels in the placenta. In addition, Notch signalling has been shown to regulate the expression of VEGF receptors in endothelial cells, which is required for the proper response of endothelial cells to VEGF stimulation(4). During placenta development, TGF-beta signalling regulates the differentiation and migration of trophoblast cells, which are specialized cells that form the placenta. It also regulates the remodelling of maternal blood vessels that supply nutrients and oxygen to the developing fetus. TGF-beta works in conjunction with VEGF to regulate angiogenesis and blood vessel formation in the placenta (5). The exact interactions of these cytokines, immune cells, metabolites/amino acids, trophoblast function, and stromal cells, and angiogenic factors are not well defined in the literature. Hence, it is simplistic to attribute the cause of idiopathic RM to one angiogenic factor family, VEGFs. Future studies could attempt to explore the role of VEGF amongst other angiogenic and immunogenic factors that have established or proposed associations with RM. “

Reviewer 2 Report

This extensive review of the possible role of the VEGF family in recurrent miscarriage is of potential interest especially to reproductive endocrine physicians and scientists. In the manuscript they compiled 13 publications of VEGFs in recurrent miscarriage. They find disparate results and ascribe the lack of a universal association to study design differences and study variables. All of this is well referenced and clearly presented. My only concern is the confusion in the introduction of which vascular system is the one studied by looking at VEGF expression. Do the authors and the 13 manuscripts attest that VEGF plays a role in both the angiogenesis in the decidua/implantation site and in the chorionic villi? I assume the former but this is not clear in the introduction and the readers of the manuscript may be confused. A small review of the normal pattern of expression during early pregnancy and a perhaps a figure showing the proposed function of VEGFs during implantation and early placental development would be most helpful. It would also be nice to include a cartoon of the uterine and placental  vasculature in early pregnancy.

Author Response

Response to Reviewer 2 Comments

This extensive review of the possible role of the VEGF family in recurrent miscarriage is of potential interest especially to reproductive endocrine physicians and scientists. In the manuscript they compiled 13 publications of VEGFs in recurrent miscarriage. They find disparate results and ascribe the lack of a universal association to study design differences and study variables. All of this is well referenced and clearly presented. My only concern is the confusion in the introduction of which vascular system is the one studied by looking at VEGF expression. Do the authors and the 13 manuscripts attest that VEGF plays a role in both the angiogenesis in the decidua/implantation site and in the chorionic villi? I assume the former but this is not clear in the introduction and the readers of the manuscript may be confused. A small review of the normal pattern of expression during early pregnancy and a perhaps a figure showing the proposed function of VEGFs during implantation and early placental development would be most helpful. It would also be nice to include a cartoon of the uterine and placental  vasculature in early pregnancy.

Thank you for your valuable feedback. We agree a figure would be beneficial. Hence we introduced figure 1.

Figure 1 The VEGF family angiogenic growth factors and receptors: interactions and functions. 

VEGF-A is a gene found on chromosome 6p12-p21.1 that codes for six different isoforms of VEGF-A, including the predominant isoform VEGF-A165, which binds to the receptors FLT-1, KDR, NRP-1, and NRP-2 and promotes angiogenesis, endothelial cell growth, and vascular permeability. PlGF, which has 42% amino acid sequence identity with VEGF-A, is predominantly expressed in the placenta, heart, and lungs, and plays a role in regulating VEGF-dependent angiogenesis under pathological conditions. VEGF-B, which forms stable heterodimers with VEGF-A, binds to the receptors FLT-1 and NRP-1 and behaves as an endothelial cell mitogen. VEGF-C regulates the lymphatic system during embryogenesis and adult life, while VEGF-D plays a role in lymphangiogenesis and can also activate the receptors FLT-4 and KDR. FLT-1 and KDR are receptor tyrosine kinases expressed on vascular endothelial cells that bind to VEGF-A, VEGF-B, and PlGF, while NRP-1 and NRP-2 are co-receptors that enhance the binding of VEGF-A and PlGF to their respective receptors.

Round 2

Reviewer 1 Report

Many thanks to authors for their responses.

For meta-analysis individual patient data is not always needed but means reported in the papers can be used. Particularly for serum VEGF there are at least five studies (Refs 17, 21, 22, 29, 30) that can be combined and meta-analysis performed. I can understand for VEGF expression using IHC it would be challenging to do so. The inability to perform meta-analysis should be discussed as a limitation of the study.

Also, information on FKBPL is not entirely correct. FKBPL does not regulate VEGF, in fact it was shown that it works via CD44 (doi: 10.1371/journal.pone.0055075 & doi:10.1158/1078-0432.CCR-10-2241). Although there might be a correlation this is indirect. This needs to be rectified. 

Author Response

Thank you for your comments.

The following has been adjusted as response to your comments.

The Student’s Independent Samples t-test was conducted using Jamovi Version 2.3.21.0 to test the null hypothesis that the serum VEGF levels (pg/L) is the same between cases of RM and control (36).

Serum VEGF data were exported into Review Manager 5.4.1 for quantitative data analysis. Metanalysis command with random effect to account for heterogeneity was used to estimate the mean difference between serum VEGF in RM cases compared to control. A quantification of heterogeneity across studies was presented as an I2 score (37).

Almawi et al., were excluded in the meta-analysis and the Student’s Independent Samples t-test as median values were reported and no raw data were available to calculate the mean (29)

The results were demonstrated in Figure 3 and 4 and explained in the results as follows:

An Independent T-test showed no evidence that the mean serum VEGF level (pg/mL) in cases with recurrent miscarriage is different from the control. The mean serum VEGF levels (pg/mL) in control was 160.94 pg/mL compared to the mean serum VEGF levels in cases 148.66 (t(8) = 0.21, p = 0.84). (Figure 3).

Upon meta-analysis, the overall mean difference between serum VEGF was 12.71pg/ml. Levels were higher in controls compared to cases of RM (confidence interval [CI] 95% -58.7 - 33.27%, I2:93%). This data was not statistically significant and had a high level of heterogeneity. (Figure 4)

The following paragraphs were added in the conclusion

To explore the association between VEGFs and RM, it is worth noting that multiple angiogenic and immunogenic factors influence endometrial and placental tissue. There are key angiogenesis pathways that could be contributing to this process and overall angiogenic balance of the placenta including TGF-beta (Tumour Growth Factor Beta), FKBPL (FK506-binding protein like), CD44, HSP90 (Heat shock protein 90), and Notch. In the context of placenta development, CD44 and VEGF may be involved in a complex signalling network that regulates placental development. Specifically, CD44 has been shown to modulate the expression and activity of VEGFR1 and its ligands, such as VEGF, in trophoblasts. CD44 has also been shown to regulate trophoblast migration and invasion by modulating VEGF signalling. Moreover, CD44 has been suggested to function as a co-receptor for VEGFR2, another VEGF receptor, and to enhance VEGF-mediated signalling in endothelial cells. This suggests that CD44 may play a role in the formation and maintenance of the placental vasculature (30). FK506 inhibits migration of human microvascular endothelial cells through binding to the CD44 receptor, leading to downstream effects on the actin cytoskeleton and alterations in expression of CD44 and related proteins, suggesting a co-regulatory pathway between FKBPL and CD44 (31, 32). In placenta development, HSP90 has been shown to interact with and stabilize VEGF, allowing for proper VEGF signalling and angiogenesis. HSP90 also plays a role in regulating the expression of VEGF receptors, such as VEGFR2, on endothelial cells. This interaction between HSP90 and VEGF is important for the development and maintenance of the fetal vasculature and ensuring adequate blood flow (33). Studies have shown that VEGF induces the expression of Notch signalling components in endothelial cells, which, in turn, promotes the formation of new blood vessels in the placenta. In addition, Notch signalling has been shown to regulate the expression of VEGF receptors in endothelial cells, which is required for the proper response of endothelial cells to VEGF stimulation (34, 35). During placenta development, TGF-beta signalling regulates the differentiation and migration of trophoblast cells, which are specialized cells that form the placenta. It also regulates the remodelling of maternal blood vessels that supply nutrients and oxygen to the developing fetus (34, 35). TGF-beta works in conjunction with VEGF to regulate angiogenesis and blood vessel formation in the placenta (34, 35). The exact interactions of these cytokines, immune cells and angiogenic factors are not well defined in the literature. Hence, it is simplistic to attribute the cause of idiopathic RM to one angiogenic factor family, VEGFs. Future studies could attempt to explore the role of VEGF amongst other angiogenic and immunogenic factors that have established or proposed associations with RM.  

Finally, this systematic review had limitations in conducting a meta-analysis. One study calculated the median serum VEGF levels as opposed to the mean and hence was excluded from the quantitative analysis (29). Some studies investigated VEGF and others specifically investigated VEGF-A or VEGF-C and hence a quantitative comparison is not completely accurate. In terms of measuring levels of VEGF using IHC, most studies did not include raw numbers of cases and controls that fell into each category of strength and intensity of staining which made it difficult to quantitively analyse the data. The heterogeneity was high, therefore, some other confounders such as population characteristics within each study need to be assessed to potentially explain the variability found in this meta-analysis. Ideally, individual participant data (IPD) meta-analysis is needed to get more accurate estimates. Despite these limitations, our results yield important conclusions and approaches to assess VEGF levels in future studies.

Furthermore, the review underwent extensive English review.

Kind regards,

Nadine Abu-Ghazaleh
